**Determining the depth and upwelling speed of the equatorial Ekman layer from**
**surface drifter trajectories**
Nathan Paldor[1] and Yair De-Leon[1]
[1]Fredy and Nadine Herrmann Institute of Earth Sciences
Hebrew University of Jerusalem
Edmond J. Safra Campus, Givat Ram, Jerusalem, 9190401 Israel

13 Correspondence to: N. Paldor (nathan.paldor@mail.huji.ac.il)

**Abstract**
Trajectories of more than 500 drogued surface drifters launched since 1979 in the equatorial ocean are
analyzed by employing the results of a new Lagrangian theory of the poleward transport from the
equator forced by the prevailing Trade winds. The Lagrangian theory provides an explicit expression for
the depth of the Ekman layer that circumvents the application of the 3-Dimensional continuity equation
that requires calculating the divergence of horizontal transport, which was the basis of all previous
studies on the subject. The analysis is carried out for drifters launched within 1° of the equator that
reached final latitude of 3°, 4° or 5° North or South of the equator while remaining in one hemisphere
throughout the entire travel time. The analysis yields robust estimates of 45 meters for the Ekman layer's
depth and 1.0 m/day for the upwelling speed of thermocline water into the layer.

## 1. Introduction

The trade winds that blow westward in the Tropics due to action of the Coriolis force on the northerly surface winds of the Hadley circulation are the first and basic component of the heat transport from the warm equatorial surface ocean to the cold poles that mitigates the overall pole-to-equator temperature gradient on Earth. The mechanism that enables the poleward time-independent heat transport is the surface flow in the ocean that is directed 90° to the right/left of the wind in the northern/southern hemisphere relative to the direction of the overlying wind. This counter-intuitive flow direction results from Earth's rotation that adds the Coriolis force to the stress applied by the winds at the ocean surface. This straightforward scenario of wind-driven ocean circulation appears in all textbooks (Knauss, 1996; Talley et al., 2011) but despite its convincing simplicity, currently, quantitative estimates of the parameters that control it vary widely depending on the data and method used in the calculations of these estimates.

The classical theory that describes the ocean response to forcing by the overlying winds was developed about 120 years ago by V.W. Ekman under the assumption of constant Coriolis frequency (Ekman, 1905). This assumption greatly simplifies the analysis by ensuring that all coefficients in the governing equations are constant. The dynamics described in Ekman's theory includes a steady flow perpendicular to the wind direction and inertial, i.e. force-free, oscillations at the local (constant) Coriolis frequency. This is in sharp contrast to the equatorial region where the Coriolis frequency vanishes at the equator and varies (linearly) with latitude, which turns the equations nonlinear so the oscillation-free flow is not steady as in Ekman's original mid-latitude theory. The poleward directed surface flows in both hemispheres along the equator imply a strong horizontal divergence along the equator which can only be balanced by the upwelling of deeper water into the wind-forced, Ekman, layer. Though the heuristic application of the mid-latitude Ekman theory to the vicinity of the equator is quite straightforward, to-

date, it could not be employed to estimate either the depth of the equatorial Ekman layer or the rate of
upwelled volume of water.
The complications that result from the inclusion of the meridional variation of the Coriolis frequency in
Ekman's theory were recently resolved in a theory of wind-driven flow in which Ekman's 1905 classical
theory was extended to the equatorial region (Paldor, 2024). This new theory employs the adiabaticity
method (Goldstein, 1980; Paldor and Friedland, 2023) that filters out fast oscillatory dynamics from the
slow dynamics associated with the motion of the center of oscillation. In the context of the equatorial
Ekman problem the eliminated oscillations result from the meridionally varying Coriolis force while the
slow and monotonic poleward motion results from the combination of the wind stress and the Coriolis
force. The essence of the method is the formulation of the problem as the dynamics of a (quasi-)particle
about the minimum of a potential while the potential itself varies with time on a slower time scale than
the period of oscillations about the minimum. The potential is derived from the meridional Lagrangian
momentum equation when the zonal velocity, $U$, is expressed as $U = D + \int f(y)dy$ where $f(y)$ is the
latitude-dependent Coriolis frequency $(= \beta y)$ and $D$ is the pseudo angular momentum, which is
conserved in the absence of other (body) forces. The substitution of the angular momentum for the zonal
velocity is essential for the analysis in spherical coordinates RomKedar et al. (1997). Additional details of
the theory are given in Sect. 2.1.
Direct observations of the depth (thickness) of the equatorial Ekman layer and the rate of upwelling
water to it are difficult to quantify due to the poor observational definition of the layer's dimensions and
the extremely low speed of upwelling. Halpern and Freitag (1987) and Johnson et al. (2001) estimated the
upwelling rate in the equatorial Pacific Ocean above 50 m depth to be about 2 m/day from the
divergence of several moored horizontal current meters. Below 50 m depth their estimated vertical
velocity is negative (directed downward). A similar upwelling rate of 2 m/day extending to depths of 120
m was estimated by Halpern et al. (1989) between 110°W and 140°W from December 1983 – September
1984 (but excluding April 1984) using the same method of inferring vertical speeds from the divergence
of horizontal currents measured by moored current meters. The upwelling speed decreased eastward in
these observations and the variation of the observed values greatly exceeded the mean values. In the
same region (Central Pacific) but in February 1980 − March 1980 Bubnov (1989) used a similar method of
integrating the 3D continuity equation associated with observed horizontal currents, and estimated the
upwelling velocity over the upper 300 m to vary between 1 and 8 m/day.
Surface drifter trajectories deployed in the Eastern Pacific during 1977-1982 were used by Hansen and
Paul (1987) as proxies of the currents and the trajectory divergence as a proxy of the current's
divergence, which yielded an upwelling speed of 1.5 m/day in a stripe of $\pm 1.5°$ of the equator. The idea
underlying their analysis is that the drifter trajectories represent the currents and divergence in the top
50 m. As noted by the authors, the main issue with their analysis is the accuracy of representation of
currents by drifter trajectories. Using over 700 drifters launched between 1979 and 1990 Poulain (1993)
estimated an upwelling velocity of $15 − 20$ m/day between $90°W$ and $150°W$ in the equatorial Pacific.
The method used by Poulain (1993) in the interpretation of drifter observations is to average the drifter
velocities in given geographical domains and at selected time intervals to generate the Eulerian velocities
at the center of the domain at that time. The high values of upwelling velocity in this study result from
the small areas used for inverting the observed Lagrangian drifter velocities to Eulerian fields. Two
important conclusions emerge from that study: The first is that except for its western part, the horizontal
divergence in the equatorial Pacific is quite uniform and therefore so should be the upwelling speed. The
second is that the upwelling velocity decreases monotonically with the assumed width of the meridional
band over which the horizontal divergence is calculated.
Estimates of upwelling rates were also calculated based on the distribution of $^{14}$C released in large
amounts to the atmosphere between 1955 and 1963 when above-ground nuclear tests took place. An
analysis of the oceanic uptake of $^{14}$C and its redistribution in the Pacific Ocean carried out by Quay et al.
(1983) yielded a value of about 0.3 m/day for the upwelling velocity along the equator. Aside from this
geo-isotopic study the previous estimations of the horizontal divergence fields were obtained either
directly from current meter observations or indirectly from drifter trajectories. In the latter case that data
were either spatially averaged to yield the Eulerian fields or interpreted as proxies of these fields. In
contrast, the current study applies a recently developed Lagrangian dynamical theory directly to observed
drifter trajectories which bypasses the need to estimate first the horizontal divergence. The large number
of drifters, the accurate tracking of their location by satellites and the longtime of coverage allows for a
selection of sufficient number of drifter trajectories that satisfy pre-determined selection criteria and
yields accurate estimates of the depth of the equatorial Ekman layer and the speed of upwelling into it.
The application of the Lagrangian theory to drifter observations and the data used in this study are
detailed in Section 2. In Section 3 we give the results obtained by applying the theory to drifter
trajectories and the study is summarized in Section 4.
**2. Theory and Data**
**2.1. Theory**
The recent extension of the wind-driven theory of ocean circulation to the equator described in Paldor
(2024) has demonstrated that, as in Ekman's original theory, the oceanic response can be decomposed
into a monotonic, slow, flow (which is directed poleward in the equatorial region) and fast, large
amplitude, oscillations. In contrast to Ekman's original theory, in the equatorial region when the wind
stress is directed westward the oscillations are highly nonlinear and of large amplitude. An example of the
large amplitude inertial oscillations associated with an initial impulse of a westward directed velocity
when no wind stress affects the motion is shown in Fig. 1a. The present study applies the explicit
expressions developed in Paldor (2024) to the observed trajectories of surface drifters and employs the
nondimensional variables and parameters in Eq. 10 of Paldor (2024). The dimensional counterpart of this
equation is:

$$\frac{dy}{dt} = \frac{1}{y(t)} \frac{-\tau^x}{H} \left(\frac{R_e}{2\Omega\rho}\right).$$ (1)

Here, y(t) is the distance from the equator at time $t$. The global parameters in this relation are: $\rho =$
$1027 \text{ kg m}^{-3}$ (water density), $\Omega = 7.29 \cdot 10^{-5} \text{ s}^{-1}$ and $R_e = 6371 \cdot 10^3$ m (Earth's rotation frequency
and radius, respectively) so $\left(\frac{R_e}{2\Omega\rho}\right) = 4.25 \cdot 10^7 \text{ m}^4 \text{ s kg}^{-1}$. The remaining, particular, parameters are: $\tau^x$
- the wind stress (units: $N \text{ m}^{-2}$; negative for easterly winds) and $H$ (m) - the Ekman layer's depth.
Multiplying the nonlinear relation (1) by $y(t)$ and integrating the resulting 1$^{st}$ order equation for $y(t)^2$
yields:

$$y(t)^2 = y(0)^2 + 2\frac{-\tau^x}{H} \cdot \left(\frac{R_e}{2\Omega\rho}\right) t.$$ (2)

As demonstrated in Fig. 1, this expression (shown by the red curve) successfully filters out the inertial
oscillations from the actual latitude time-series (blue curves), and describes the net, oscillation-free,
poleward motion.
Inverting Eq. (2) to an explicit expression for $H$, setting $t$ to $t_i$, the travel time of drifter #$i$, and $y(t)$ to
$L$, a "boundary" of the equatorial region, yields the estimate of $H_i$ the depth value based on trajectory #$i$:

$$H_i = \left(\frac{R_e}{2\Omega\rho}\right) \frac{2(-\tau^x)}{L^2 - y_i(0)^2} t_i,$$ (3)

where $y_i(0)$ is the distance of drifter #$i$ from the equator at $t = 0$ (i.e. the distance of the launch point
from the equator).
**2.2. Drifter trajectories**
Nearly 30,000 surface drifters were released from 1979 at the ocean surface (Lumpkin et al., 2017) and
the geographical trajectories of these drifters are tracked by satellites every 6 hours for periods of up to
1000 days. These (Lagrangian) observations cover the global ocean and a few percent of them were
launched on both sides of the equator in the Pacific, Atlantic and Indian oceans. The slightly negatively
buoyant drifter is typically drogued at 15-meter depth, so it provides an estimate of the current in the top
15 meters of the water column where the wind stress is the primary forcing (Lumpkin et al., 2017). The
agreement between drifter trajectories and ocean currents demonstrated in Lagerloef et al., (1999) and
assumed in Poulain (1993) motivates an analysis of observed drifter trajectories in order to determine the
depth of the equatorial Ekman layer and the upwelling speed of deep water into it.

The drifter trajectories used in the analysis are freely available from NOAA Global Drifter Program

(NOAA/AOML/GDP) site. The data were screened according to the following three criteria:
1. They were launched within $1° \approx 110$ km south or north of the equator (regarded as the equator).
2. Once launched, the drifters remained in one hemisphere throughout the entire travel time to the

final latitude (this is because equator crossing is not possible under the assumed westward directed

wind stress).

3. The drifters were continuously tracked, with gaps no longer than one day, during their motion from

the launch point to the final latitude that marks the boundary of the equatorial region (i.e. $3° \approx$

330 km, $4° \approx 440$ km or $5° \approx 550$ km).

The latitudes $3°$ and $4°$ were used in previous studies to define the boundaries of the equatorial region
(Brady and Bryden, 1987; Lagerloef et al., 1999; Johnson et al., 2001) but in the present study we also
used $L = 5° \approx 550$ km to verify the robustness of the calculated averages to the selected values of $L$.
The case $L = 2°$ is not included in the analysis since the singularity of Eq. (3) at $L^2 = y_i(0)^2$ affects the
accuracy of the estimates of $H_i$ when $L$ is close to $y_i(0)$ i.e. for $L = 2° \approx 220$ km and for $y_i(0) \leq 1°$ the
denominator is tiny which can yield extremely high value of $H_i$ and amplify observational errors.

As of August 2024 ~30,000 drifter trajectories are archived in AOML archive and over 1500 drifters

were launched near the equator and reached the final latitudes of $3°, 4°$ or $5°$, out of which ~700 drifters
remained in one hemisphere. The number of drifters in the Atlantic and Pacific oceans that reached each
of the final latitudes is given in the 2nd column of Table 1 that also gives the mean $y_i(0)$ ($\equiv Y(0)$, 3rd
column) and the mean $t_i$, ($\equiv T$, 4th column) to the final latitudes (noted in the rows of this table). The
Indian Ocean is excluded from the analysis due to the positive mean annual wind stress in it (see Sect.
2.3). No screening was made of drifters that lost their drogues on their way from $y_i(0)$ to $L$.
**2.3. Wind stress**
The daily wind stress values over the oceans, $\tau^x$, used in this work are available at NOAA/CoastWatch site
in $0.125°$ spatial resolution for $1999 - 2009$. We calculated the averaged wind stress for the whole period
in the entire region of the Indian, Atlantic and Pacific oceans in a zonal strip that straddles the equator
between $-L$ and $+L$, where $L$ corresponds to $3°$, $4°$ or $5°$.

| $L$ (degrees) | Number of drifters | Mean $y_i(0)$ (degrees) | Mean $t_i$ (days) | $\tau^x$ $(Nm^{-2})$ | Mean $H_i$ (m) | $W = \frac{H}{T} \cdot \frac{L-Y(0)}{L}$ (m/day) |
|---|---|---|---|---|---|---|
| 3.04 | 610 | 0.29 | 29.67 | -0.0261 | 51.10 | 1.56 |
| 4.04 | 576 | 0.29 | 43.43 | -0.0264 | 42.48 | 0.91 |
| 5.04 | 531 | 0.29 | 58.12 | -0.027 | 37.16 | 0.60 |


**Table 1: Drifter characteristics in the Pacific and Atlantic oceans and the zonal wind stresses there. The**
**shown values of $L$ are larger by a few kilometers compared to the distances corresponding to $3°$, $4°$ or**
**$5°$ since a drifter is determined to be "at $L$" with an offset of up to 6 hours after its passage of that**
**point. Less than 10% percent of the relevant drifters were launched in the Indian ocean which is not**
**included in this table and in the analysis since the annual mean wind stress in the Indian Ocean is**
**directed eastward, which is inconsistent with a poleward directed net motion. The 5th column denotes**
**the mean wind stress daily values over the entire $1999 - 2009$ period in each ocean used in this study.**
**The variables $H, T$ and $Y(0)$ denote the mean values (over all drifters) of $H_i, t_i$ and $y_i(0)$, respectively.**
These wind stress averages are given in the 5th column of Table 1 for the Atlantic and Pacific oceans,
where the values are nearly identical (they differ by no more than a few percentage points) but not for
the Indian ocean where the calculated mean values of the wind stress are positive (so $H_i < 0$) and small
(less than $+0.01$ N m$^{-2}$) probably due to the strong seasonal forcing by the Monsoon system that
induces eastward directed zonal winds throughout part of the year in this ocean (Hastenrath and Polzin,
2004; Zhang et al., 2022). The decade-long zonal wind stress observations are considered representative
of the climatic values that prevailed throughout the trajectories of all drifters. Though the negative mean
temporal values of $\tau^x$ in the Atlantic and Pacific Oceans are not spatially uniform (reaching their maximal
values in the center of each ocean and tapering off near the continents that bound the ocean on the east
and west sides) only the spatially mean values are used.

## 3. Results

Four representative drifter trajectories are shown in Fig. 2 and they demonstrate the richness of observed
trajectories near the equator, the intricate combination of oscillations with slow poleward propagation
and the occurrence of equatorial crossing in many trajectories.
Substituting the values of $y_i(0)$ and $t_i$ for each drifter in Eq. (3) yields the corresponding values of $H_i$.
The 6$^{\text{th}}$ column of Table 1 gives $H$, the mean of the particular values of $H_i$, and the histograms of the $H_i$
values for each value of $L$ are shown in Fig. 3 so the value of $H$ is best estimated by: $H = 44 \pm 7 \approx 45$ m.
Equation (2) can also be employed to calculate the poleward, oscillation-free, velocity of a drifter on
its way from $y_i(0)$ to $L = y(t_i)$ from the drifter's average speed during its travel: $V_i = \frac{L - y_i(0)}{t_i}$. Thus, the
volume divergence (per unit length in $x$) that results from the anti-parallel, poleward directed, volume
fluxes of 2 water columns that are initially conjoined along the equator and move poleward is $2H_iV_i =$
$2H_i \frac{L - y_i(0)}{t_i}$. The vertical volume flux (per unit length) due to Ekman upwelling during $t_i$ is: $2LW_i$, where
$W_i$ is the upwelling speed. Equating the vertical and horizontal fluxes yields $W_iL = H_i \frac{L - y_i(0)}{t_i}$ or $W_i =$
$\frac{H_i}{t_i} \frac{L - y_i(0)}{L}$. The mean values of $H_i$ and $t_i$ in Table 1, denoted by $H$ and T, then yield the mean values of $W$
in the 6$^{\text{th}}$ column that averages to $W \approx 1.0$ m/$day$ for the three values of $L$.
The estimated $H$ and $W$ values along the equatorial Atlantic and Pacific Oceans are noted in Fig. 4 on a
qualitative, textbook, cartoon of the wind forcing and resulting oceanic flow patterns.

## 4. Summary and Discussion

The mean estimates of $H = 45\ m$ and $W = 1\ m/day$ calculated here based on surface drifter
trajectories are more robust compared to prior estimates derived from standard hydrographic
observations. Table 1 shows that the present estimates of $H$ vary with $L$ by a few meters and those for $W$
by about 0.5 $m/day$. These variations are smaller than those of estimates based on standard
hydrographic data that can vary by about one order of magnitude (Wyrtki, 1981; Brady and Bryden, 1987;
Lukas and Lindstrom, 1991; Weingartner and Weisberg, 1991). As an example, the $H = 45\ m$ value
reported here exceeds the estimate of 30-40 m proposed in Lukas and Lindstrom (1991) but the $O(10\%)$
variation of the present estimate is significantly smaller than the $O(80\%)$ variation in the previous
estimate. In view of the crucial role played by the poleward flow of warm equatorial water in mitigating
the large radiative pole-to-equator temperature gradient (Czaja and Marshal, 2006; Hartmann, 2016) a
reliable quantification of the initiation of this flow is important for understanding Earth's climate.
The zonal stress is not uniform, reaching a maximal value in the center of the Ocean and tapering off
towards the boundaries on the east and west. The effect of this variation on the value of $H$ Is pronounced
in the equatorial Pacific Ocean and calculations of $H$ using the values of $\tau^x$ at each drifter's initial location,
yield a wider range of $H$ values. These estimates, too, are not exact since the value of $\tau^x$ varies with time
and along the drifter trajectory. A detailed analysis of the variation of $\tau^x$ along a drifter trajectory and its
effect on the value of $H_i$ calculated in that trajectory are left for future work.
In contrast to Ekman's theory in mid-latitude, the extension of this theory to the equatorial region
implies that the slow net poleward motion of a water column subject to a zonal wind stress is
accompanied by a zonal component. This results naturally from the inertial oscillations that involve net
zonal translation on the equator (as in Fig. 1a), while inertial oscillations on the mid-latitudes $f$-plane are
not accompanied by a translatory motion.
The results calculated in Sect. 3 focus on the net poleward propagation rate, which neglects the
inertial oscillations, while the observed trajectories include both types of motion. The neglect of inertial
oscillations in the observed trajectories is justified based on the fact that the period of these oscillations is
much shorter than the temporal length of the averaged trajectory used for calculating $H_i$ in Eq. (3). The
high variability of the geographical trajectories, which is exemplified in Fig. 2, does not permit an estimate
of the oscillations' period directly from the trajectories. However, for the values of $\tau^x = 0.026 \ Nm^{-2}$ and
$H = 45 \ m$ the period of inertial oscillations can be estimated from the scale $P = \left( \beta \cdot \frac{\tau^x}{\rho H} \right)^{-\frac{1}{3}} \approx 5 \ days$
which is much shorter than the typical 30-60 days trajectory length (see the values of mean $t_i$ in Table 1).
Thus, during the relatively long drifter travel time the poleward motion due to the oscillations averages
out to zero, leaving the poleward motion of the oscillations' centers as the sole contributor to the net
motion.
The results shown in Table 1 imply a decrease in the value $H$ (and hence the value of $W$) with the
increase in $L$. This result is not trivial since Eq. (3) is also satisfied with constant $H_i$ provided $L^2 \propto t_i$ for
$y_i(0) \ll L$. However, the calculated values of mean $t_i$ for different values of $L$ reported in Table 1 only
show a power slightly above 1.0 but significantly smaller than 2.0. The variation of $H$ with $L$ probably
originates from a mechanism similar to that depicted by the blue arrows in Fig. 4b where the depth of the
Ekman layer becomes shallower with the distance from the equator, which affects the relation between T
and $L$. The suggested mechanism is reminiscent of the thinning of smoke billows far from the smoke stack
but we note that the transition from Eq. (1) to Eq. (2) is valid only when $H$ is independent of $t$ and $y$.
Though no explanation is proposed here for the quantitative dependence of $H$ on $L$ this result is
consistent with the results reported in Fig. 3 of Poulain (1993), which show that the horizontal divergence
decreases monotonically with the width of the latitudinal band. We also note that Eulerian calculations of
$\frac{\partial v}{\partial y}$ based on spatial averaging of the Lagrangian observations as was done in Poulain (1993) yield an
estimate of $W/H$ so additional information is needed for determining each of these parameters (Poulain,
1993 arbitrarily assumed $H = 50\ m$). In contrast, in the present study $H$ is determined directly from
drifter trajectories, which yields an estimate of $W$ based on mass conservation in a box of meridional
extent 2*L*, thickness *H* and unit zonal length.

The successful application of the new dynamical theory of wind-driven equatorial transport in the

ocean developed in Paldor (2024) lends credence to the relevance of this theory to observations in the
equatorial ocean and in particular to the use of the velocity at 15 m as representative of the surface
Ekman layer. This relevance is demonstrated despite the omission of important factors such as the
meridional wind stress (can be significant at short times), the initial drifter velocity (assumed to vanish in
the theory) or the spatial and temporal changes of the zonal wind stress. It can be argued that the
calculation and successful application of the oscillation-free speed of poleward wind-driven motion on
the equator is of similar significance to ocean dynamics as the development of the expression for the
steady transport in the original *f*-plane Ekman theory.

**Author contribution**:
NP: Initiation of project, writing various drafts and theoretical analysis.
YD: Data collection and analysis, editing and production of display items.
**Competing interests**: The authors declare that they have no conflict of interest.
**Financial support**: The authors happily declare that no funding was received for this study
**Data Availability**: The drifter data used in this study are maintained by NOAA/AOML and the
wind stress data are maintained by NOAA/CoastWatch, both sites are noted in the list of
references
**Acknowledgements:** The authors are grateful to two anonymous reviewers whose comments
greatly improved the presentation of this paper.

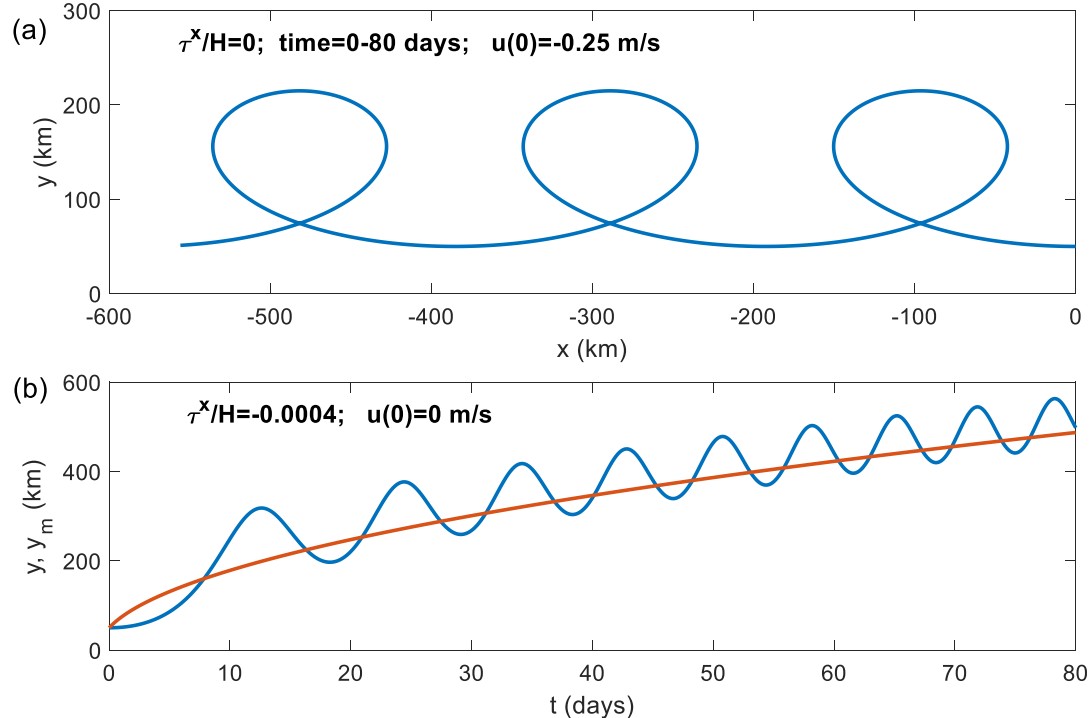

**Figure 1: Panel (a): The numerically calculated geographic trajectory of a water column initiated**
**with a westward directed impulse of 0.25 m s$^{-1}$ zonal velocity subject only to the, latitude-**
**dependent, Coriolis force. Panel (b): the latitude (i.e. distance from the equator in km) time-series**
**(blue curve) and the oscillation free latitude time-series given by Eq. (2) denoted here by $y_m$ (red**
**curve) of a 50 m deep water column forced by a westward directed wind stress, $\tau^x$, of 0.02 N m$^{-2}$. In**
**both panels the initial distance from the equator is $y(0)=50$ km while the initial longitude and the**
**initial meridional velocity are both 0.**

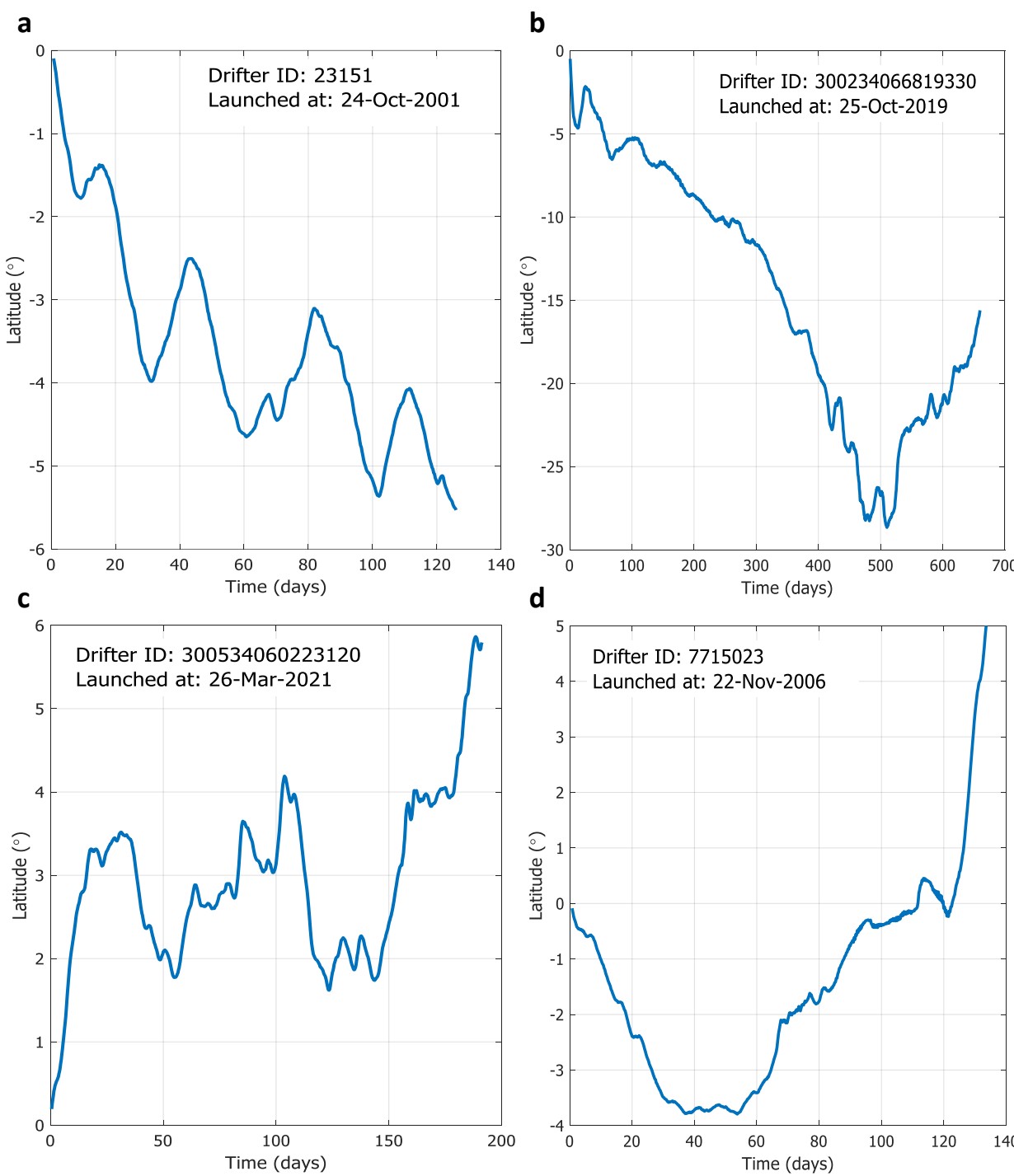

Figure 2: Four drifter trajectories originating within $1°$ of the equator analyzed in this study. a) A typical southern hemisphere trajectory that clearly shows oscillations and a mean poleward flow; b) A fast southern hemisphere trajectory that reaches $4°$ in just a few days and remains operational for nearly 2 years; c) A slow northern hemisphere trajectory that reaches $4°$ in more than 100 days; d) Part of a trajectory that reaches $3°$ prior to crossing the equator so it is included in the analysis of $L = 3°$ but not in the $L = 4°$ or $L = 5°$ analyses since it reaches these latitudes only after crossing the equator.

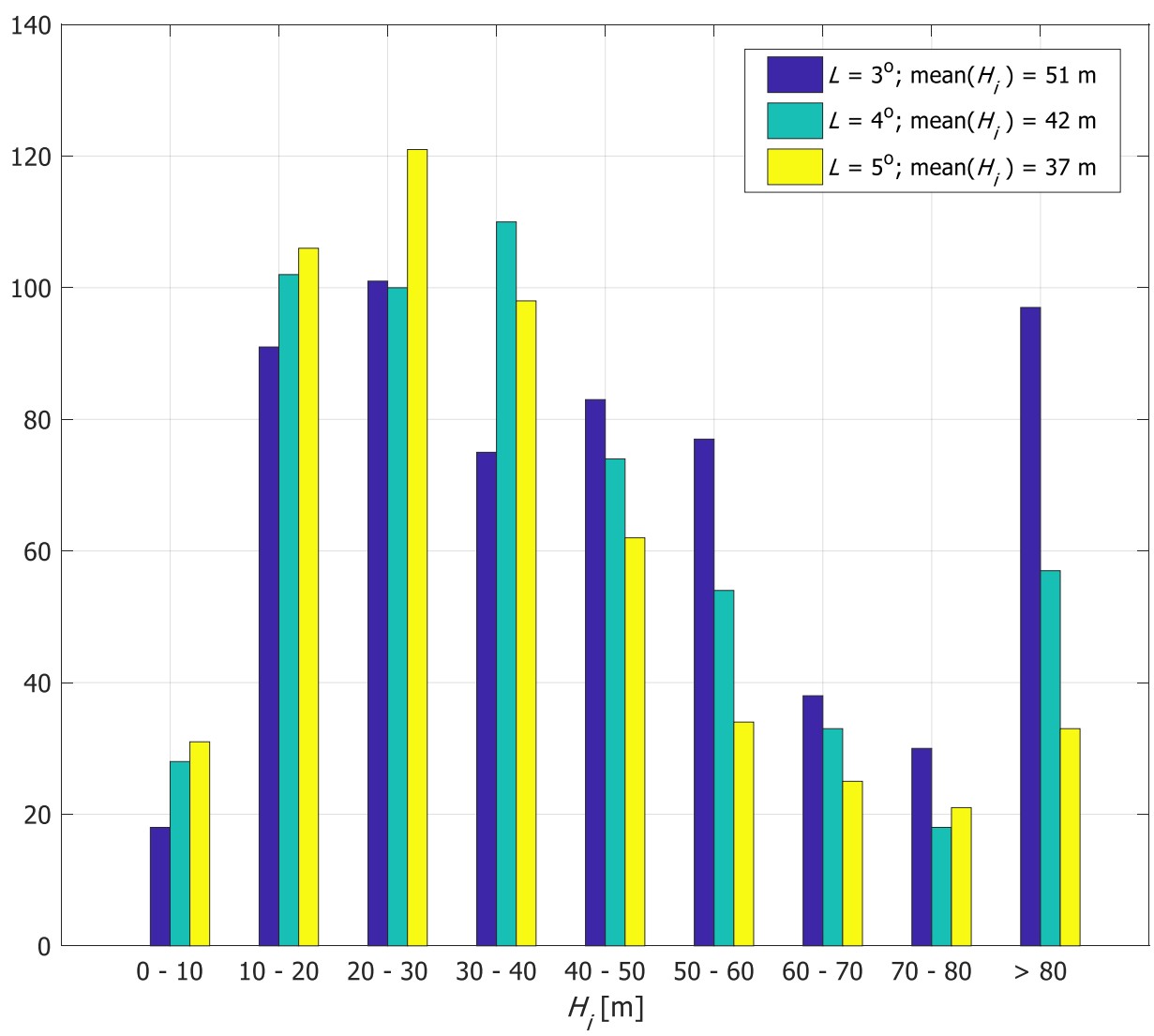


**Figure 3: The histograms of $H_i$-values for the 3 values of $L$. For $L = 3°$ the tail of $H_i > 80$ m is as high**
**as the maximum cell of $H_i = 20 - 30$ m consistent with singularity of Eq. (3) at $L = y_i(0)$.**

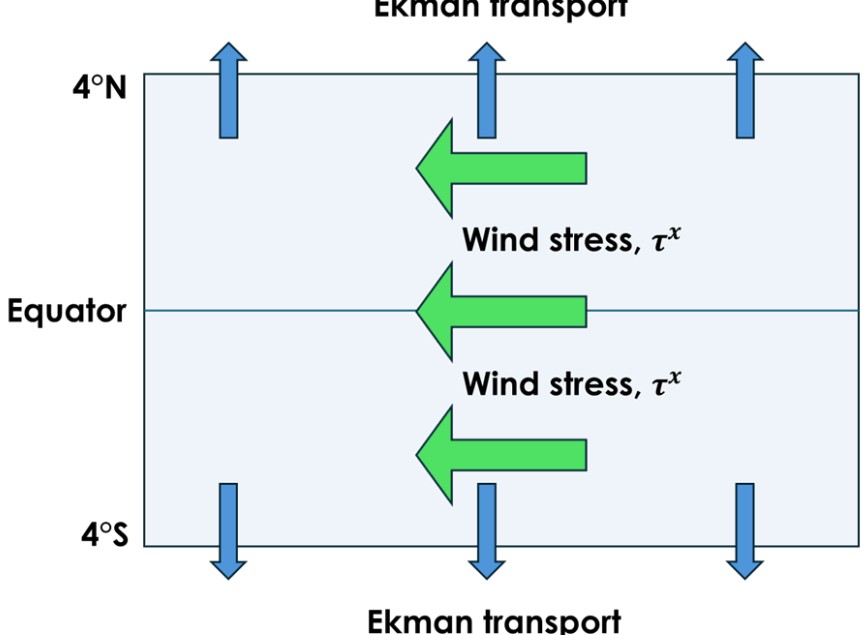

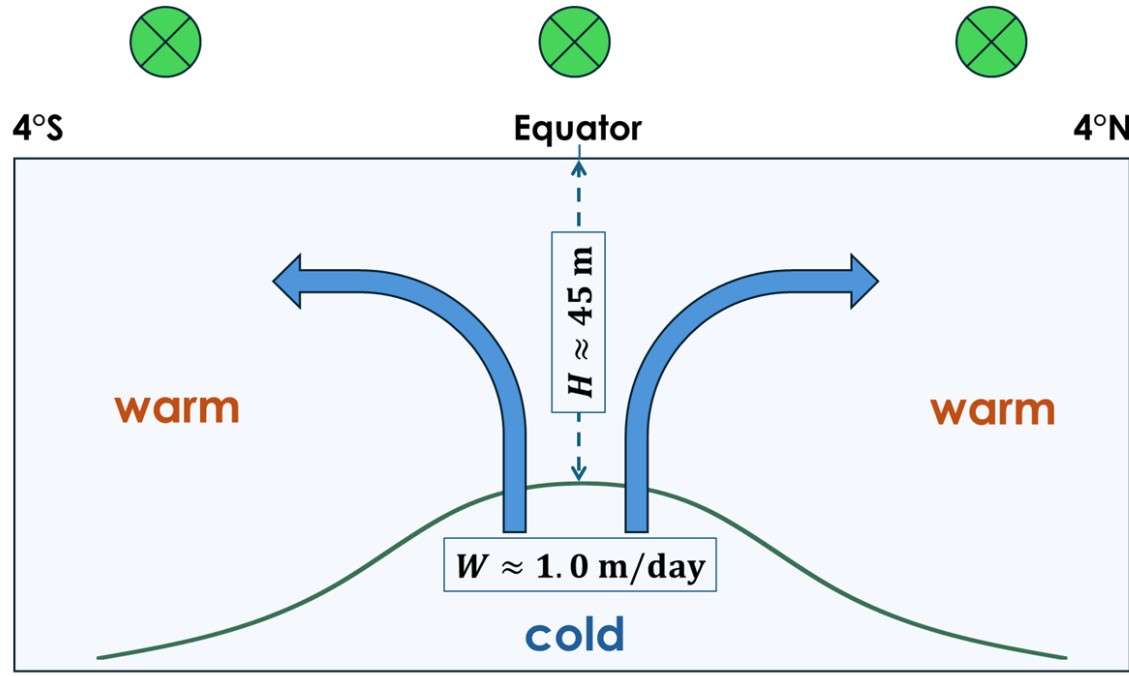


Figure 4: A sketch relating the poleward directed wind-driven surface flow along the equator under westward directed wind stress (upper panel – planar view) which is compensated by the upwelling of water from below (lower panel – latitude-height cross-section viewed from the east). The dark green curve in the lower panel denotes the boundary between the warm surface water and cold thermocline water. The $H \approx 45$ m and $W \approx 1.0$ m/day estimates are the main results of this study.

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
