# Peer review of "surface drifter trajectories"

_EGUsphere, 2025_

## Referee Comment (RC1)

**Reviewer Comments**

**General Comments:**

The classical Ekman theory primarily focuses on the characteristics of Ekman currents in mid- and high-latitude regions. This manuscript investigates Ekman layer depth and Ekman plumping speed in the equatorial region using surface drifter trajectories. This approach demonstrates a certain level of innovation. The manuscript would be better from a more detailed explanation of the methodology and assumptions used in the calculations. Since some readers may not have read the authors' previous paper: Paldor (2024), it would be helpful to provide an overview or reference to the key findings or methodologies from that work to ensure clarity and context for the current study. This will help readers better understand the foundation and progression of the research presented in this manuscript. Additionally, the discrepancies in the data ranges need to be addressed to improve the overall quality of the manuscript.

Here are some specific suggestions:

**Lines 18-22:** Please provide more details in your abstract. For example, the specific latitude defining the equatorial ocean should be clarified. Specifically, what are the deflection angles between current velocity and wind speed, as well as between current velocity and water mass transport? It should be explained why the angle is 90°. How are the equatorial current and the equatorial Ekman flow distinguished? Why it is defined as equatorial Ekman flow?

**Line 43:** There is an extra comma in "Ekman layer."

**Lines 50-52:** The text in these lines is difficult to comprehend and should be revised for clarity.

**Line 75:** Is the formula derived from Paldor (2024)?

**Line 91:** It is mentioned that 30,000 buoys were deployed from 1979 to 2025, but the wind stress data is from 1999 to 2009 (Line 120). This discrepancy should be clarified.

**Line 127:** Are the regional variations in the Pacific and Atlantic Oceans significant? This should be discussed in more detail.

**Line 147:** The estimation of vertical velocity w appears to be overly simplistic. A more rigorous approach, such as calculating differentials between two consecutive time points of each drifter trajectory, should be considered.

---

## Author Comment (AC1)

Response to the comments of referee #1 on "Determining the depth and pumping speed of the equatorial Ekman layer from surface drifter trajectories" (egusphere-2025-089) by Paldor and De-Leon

We were happy to read that referee's general comment that: "The approach demonstrates a certain level of innovation.". The remaining general comments are repeated in the particular comments and our detailed response to these comments are detailed follow with the comment in black and our response in blue. The comments are listed as written by the referee i.e. each comment is denoted by the line number of the original draft to which it relates.

**Lines 18-22:** Please provide more details in your abstract. For example, the specific latitude defining the equatorial ocean should be clarified. Specifically, what are the deflection angles between current velocity and wind speed, as well as between current velocity and water mass transport? It should be explained why the angle is 90°. How are the equatorial current and the equatorial Ekman flow distinguished? Why it is defined as equatorial Ekman flow?

Abstract is expanded as requested. The deflection angle is not addressed since the zonal motion of the center of oscillation results solely from the westward motion of Inertial Oscillation when the Coriolis frequency varies with y. This point requires an analysis that is beyond the scope suitable for the manuscript that only applies theoretical expressions to archived drifter observations.

**Line 43:** There is an extra comma in "Ekman layer."

Done. Thank you

**Lines 50-52:** The text in these lines is difficult to comprehend and should be revised for clarity.

A more detailed explanation of the essential theoretical background of the theory developed in Paldor (2004) now appears both in the Introduction and in Sec. 2a (see L60-67, L.117-132 and new Fig. 1)

**Line 75:** Is the formula derived from Paldor (2024)?

Yes, this is now clarified new L119-121

**Line 91:** It is mentioned that 30,000 buoys were deployed from 1979 to 2025, but the wind stress data is from 1999 to 2009 (Line 120). This discrepancy should be clarified.

The 10 yearlong wind stress is representative of the climatological winds used in Eq. (3). This length of averaged (over all relevant longitudes and latitudes) wind stress should accurately represent the climatological stress

**Line 127:** Are the regional variations in the Pacific and Atlantic Oceans significant? This should be discussed in more detail.

The regional variations are now briefly described in L220-230 (but not studied in depth as this beyond the scope of the present simple model).

**Line 147:** The estimation of vertical velocity w appears to be overly simplistic. A more rigorous approach, such as calculating differentials between two consecutive time points of each drifter trajectory, should be considered.

The method we applied is highly robust in terms of the resulting value of $H$ while the proposed method is subject to large errors at short distances due to the sensitivity of Eqs. (2) and (3) at small values of $y_i(t + \Delta t) - y_i(t)$. In addition, the suggested, short time, application does not filter out the oscillatory motion from the observed drifter trajectories. See L160-162 and L231-240 of the revised version.

---

## Author Comment (AC2)

Response to the comments of referee #2 on "Determining the depth and pumping speed of the equatorial Ekman layer from surface drifter trajectories" (egusphere-2025-089) by Paldor and De-Leon

We thank the referee for accolades in his general comments: "…This is a well-conceived and straightforward study that highlights new aspects of equatorial Ekman dynamics." and "Overall, the manuscript is well written and suitable for publication…". We also appreciate the extensive list of references included in the review and used in the greatly expanded revised version (and for which an acknowledgement was added to the manuscript).

Our detailed response to the referee's particular comments are detailed bellow with the comment in black and our response in blue. The comments are listed as in the referee's list i.e. each comment is denoted by the line number of the original draft to which it relates.

L54: Minimum potential. More information would be helpful here to make this understandable without referring directly to Paldor (2024).

A more detailed explanation of the essential theoretical background of the theory developed in Paldor (2004) now appears in the Introduction and in Sec. 2a (see L60-67, L117-132 and new Fig. 1). We feel that the repetition of previously published material (especially when published with open access) in a leading journal (Phys. Fluids) is a highly subjective issue.

L101: Did you check for drifters that lost their drouges? Should be noted.

No. A comment reflecting this appears in L168-169

L103: Typically, there is substantial meridional wind, particularly in the equatorial Atlantic. How do you account for this effect? With the selection criteria used, a significant bias could be introduced, as only drifters associated with specific non-Ekman dynamics—such as tropical instability waves, Yanai waves, or meridional wind forcing—are considered. A way to test for such a bias would be to calculate the mean meridional drifter velocity as a function of latitude. It would be interesting to see how this quantity compares to the meridional velocities derived from Ekman theory.

We agree: There can be many reasons why the simple theory adapted in the manuscript should not apply to the drifter observations. However, "the proof is in the pudding" and the robust estimates derived from the application of the theory where $H$ varies by about 10% and $W$ by 50% speak for themselves.

L111: I do not understand this sentence. Please clarify. What does it mean? L is 2° or is L approaching y(0)? Or do you mean if y(0) approches L=2°? Is the erratic behavior a consequence of meridional wind forcing?

Sentence extended and rephrased and the new sentence in L160-162 explains the point more clearly.

L113: Criterion for selection of drifters. Would it be a better criterion to consider the strength of the meridional wind, i.e., to only use drifters in cases of weak meridional wind? Additionally, what is the initial velocity of the drifters—for example, is it influenced by meridional wind forcing or, more importantly, by tropical instability waves, Yanai waves, etc.? Drifters could be deployed under different conditions.

Clearly, the conditions that prevailed when a particular drifter was launched affect its trajectory but since these conditions are not known we assume that all forces but the zonal wind stress (the value of which is negative even in the decadal mean) average out to zero for the hundreds of trajectories calculated for drifters that were launched in a period of over 40 years. Similarly, for lack of data, and as explained in Paldor (2004), the initial velocities are assumed to be 0 which agrees with the over 6-hour gap between launch time and first fix by a satellite. The important forcing of meridional wind stress can be incorporated in the dynamical theory but not as part of the present work in which the Paldor (2024) theory is applied to observations.

L140: Result of the Ekman layer depth. There must be a strong bias in the mean since you specifically neglect drifters that do not flow in the expected direction or that cross the equator. As a result, you include all drifters that may be driven by other motions away from the equator but exclude those drifting toward it.

That is correct. The theory is relevant to poleward moving drifters forced by zonal wind-stress. Thus, the "bias" alluded to in this comment is a virtue that suits the present application (see also our response to reviewer's comment L103) - the motion due to other types of forcing should be neglected.

What would be the equatorial divergence if calculated from all drifters, i.e., the mean meridional velocity averaged along the equator at different latitudes (3°, 4°, 5°)? If there is a difference, how could it be explained?

The suggested analysis was carried out in Poulain (1993) by transforming the Lagrangian data to Eulerian fields after averaging the values of $v$ in prescribed spatial bins. In the Lagrangian formulation, employed here, $\frac{\partial v}{\partial y}$ only determines the shape of the projection of the 4Dimensional solution curve onto the $(v, y)$ plane but does not imply horizontal divergence as in the Eulerian form. In contrast, the novelty of the present work is the use of Lagrangian Eq. (3) directly to circumvent the transformation from Lagrangian to Eulerian formulations. As was shown in Paldor (2024), in the Lagrangian formulation mass (height) conservation is determined by $\frac{\partial y(t)}{\partial y(0)}$. A discussion of this point now appears in L248-253.

L141: Can this be called oscillation free. How do you subtract the oscillation from the drifter velocity? To my understanding this would require the knowledge of the initial velocity.

Our method is based on the realization that the period of oscillation is (significantly) shorter than the period over which the trajectories are calculated so the oscillations are averaged out. This is explained now in L231-240.

L142: The derived velocity corresponds to the velocity at the depth of the drifters' drogue. What assumptions do you make about the vertical structure of the Ekman velocity? Would it be useful to also consider Argo drift velocities, which represent surface velocity, to calculate mean meridional velocities? This could provide an estimate of the vertical structure of the Ekman flow near the equator.

The simplified Lagrangian theory cannot be expected to decipher the 3Dimensional structure of the velocity field. As is evident from our calculations it does a reasonable job in fitting the meridional motion to drifter trajectories, which is probably due to the fact that the cation of the wind stress decays with depth so the averaged velocity over the entire 45 m deep layer is determined primarily by the velocity at the top 15 meters sampled by the drifters.

L147: I think one of the main results is the dependence of the derived vertical velocity on the meridional scale, which could also explain many previous estimates (see references below). This aspect should be given more emphasis

Thank you. The point (that agrees with the results obtained by Poulain93) is now discussed in L241-248.

---

## Referee Report (RR1)

The revised manuscript has addressed the previous concerns effectively, and the improvements have enhanced the clarity and rigor of the work. The paper is now suitable for publication, pending minor editorial revisions outlined below. With these revisions, I recommend acceptance for publication.

**Specific Recommendations**

**Line 21**: "Trade winds" should be written in lowercase as "trade winds" unless it begins a sentence.

**Line 27**: "1.0 meters/day" should follow the same style as elsewhere (lines 106, 110, 127) in the manuscript (e.g., "1.0 m/day" or "1.0 m $d^{-1}$" if metric abbreviations are preferred).

**Line 85**: "12/1983 to 9/1984" should be revised to the formal written format, "December 1983 – September 1984" (en dash), so do **Line 100**.

---

## Author Response (AR2)

Response to the second round of comments of referee #1 on "Determining the depth and pumping speed of the equatorial Ekman layer from surface drifter trajectories" (egusphere-2025-089) by Paldor and De-Leon

We Thank the reviewer for his/her careful reading of the second version of the manuscript and for the additional round of comments, all of which were implemented in the revised version.

Response to the second round of comments of referee #2 on "Determining the depth and pumping speed of the equatorial Ekman layer from surface drifter trajectories" (egusphere-2025-089) by Paldor and De-Leon

We thank the reviewer for his/her careful reading of the revised version of the manuscript and for suggesting another list of comments. Nearly all the reviewer's comments were implemented in the 3rd (current) version of the manuscript and the few exceptions are detailed below, identified by the line number (of the second version) in the list of comments and reproduced here over gray background.

L152f: I still do not fully understand this criterion. What do you do with drifters deployed at the equator, such as in Hans et al. (2024, https://doi.org/10.1029/2023JC020870), their Fig. 1? Are such drifters, deployed at the equator, included or not? Would it make sense to count all drifters that reach a position near the equator and subsequently move poleward, regardless of their prior trajectory, including whether they may have crossed the equator earlier or not?

We tracked drifters that were deployed between $1°S$ and $1°N$ so $0°$ does not pose any special case or value. We consider this

L155: launch position: Here, I would suggest to define a starting position for the calculation instead of a launch position. Anyway, taken the launch position does not yield zero meridional velocity at the trajectory start.

The initial velocities are not known and are presumably small and directed randomly in all possible directions.

L158f: No information …: It might not be used here, but the Global Drifter Program (GDP) flags the presence or absence of drogues in its dataset and information is available. Might not be the case for older drifter. See, e.g. https://doi.org/10.5194/essd-13-645-2021

We now state that we haven't screened for drifters that lost their drogues prior to reaching the final latitude

L171: If I understand correctly only mean tau_x is used instead of tau_x calculated for each drifter separately (i.e. averaged over the period of drift and longitude range) and then averaged for all drifters. This should be stated clearly right from the beginning. Which longitude ranges and periods are used for averaging?

That's correct and is now clarified in the text

L197: How is H estimated, just as an arithmetic mean? As the distribution is skewed, one could use other statistical expression such as the median. What is the difference between mean and median?

The text now shows the calculation of H for each drifter (denoted by the subscript "i" and in $t_i$ and $y_i(0)$) and then averaged over all drifters.

L237: I don't know how the given equation for the period fit to Figure 1a showing 3 oscillations in 80 days for zero wind stress.

The scale is derived for $\tau^x \neq 0$ while the oscillations in Fig. 1 are calculated for $\tau^x = 0$ (and $U(0) \neq 0$). Clearly, a finite $U(0)$ can be used to introduce another scale for the period.